# Celiac-Related Autoantibodies and IL-17A in Bulgarian Patients with Dermatitis Herpetiformis: A Cross-Sectional Study

**DOI:** 10.3390/medicina55050136

**Published:** 2019-05-15

**Authors:** Tsvetelina Velikova, Martin Shahid, Ekaterina Ivanova-Todorova, Kossara Drenovska, Kalina Tumangelova-Yuzeir, Iskra Altankova, Snejina Vassileva

**Affiliations:** 1Clinical Immunology, University Hospital Lozenetz, 1407 Sofia, Bulgaria; tsvelikova@medfac.mu-sofia.bg (T.V.); altankova@abv.bg (I.A.); 2Department of Dermatology, Faculty of Medicine, Medical University—Sofia, 1431 Sofia, Bulgaria; martin.shahidmd@gmail.com (M.S.); kosara@lycos.com (K.D.); 3Laboratory of Clinical Immunology—University Hospital St. Ivan Rilski, 1431 Sofia, Bulgaria; katty_iv@yahoo.com (E.I.-T.); kullhem000@gmail.com (K.T.-Y.)

**Keywords:** dermatitis herpetiformis, anti-tTG, anti-DGP, AAA, AGA, IL-17A

## Abstract

*Background and objectives*: Dermatitis herpetiformis (DH) is a blistering dermatosis, which shares common immunologic features with celiac disease (CD). The aim of the present study was to explore the performance of a panel of CD-related antibodies and IL-17A in Bulgarian patients with DH. *Materials and Methods:* Serum samples from 26 DH patients at mean age 53 ± 15 years and 20 healthy controls were assessed for anti-tissue transglutaminase (anti-tTG), anti-deamidated gliadin peptides (anti-DGP), anti-actin antibodies (AAA), and IL-17A by enzyme linked immuno-sorbent assay (ELISA), as well as anti-tTG, anti-gliadin (AGA), and anti-Saccharomyces cerevisiae antibodies (ASCA) using immunoblot. *Results:* The average serum levels of anti-tTG, anti-DGP, AGA, AAA, and the cytokine IL-17A were at significantly higher levels in patients with DH compared to the average levels in healthy persons which stayed below the cut-off value (*p* < 0.05). Anti-DGP and anti-tTG antibodies showed the highest diagnostic sensitivity and specificity, as well as acceptable positive and negative predictive value. None of the healthy individuals was found positive for the tested antibodies, as well as for ASCA within the DH group. All tests showed good to excellent correlations (r = 0.5 ÷ 0.9, *p* < 0.01). *Conclusions:* Although the diagnosis of DH relies on skin biopsy for histology and DIF, serologic testing of a panel of celiac-related antibodies could be employed with advantages in the diagnosing process of DH patients. Furthermore, DH patients who are positive for the investigated serologic parameters could have routine monitoring for gastrointestinal complications typical for the gluten-sensitive enteropathy.

## 1. Introduction

Dermatitis herpetiformis (DH), also known as Duhring-Brocq disease, is a rare subepidermal blistering dermatosis, currently regarded as the specific extraintestinal manifestation of celiac disease (CD) [1,2]. It most commonly affects the skin, while associated gluten sensitive enteropathy (GSE) can be clinically variable to absent. Histologically, DH is characterized by subepidermal blisters with predominant neutrophilic infiltration in the papillary dermis. A pathognomonic finding in DH, detected by direct immunofluorescence (DIF) microscopy on perilesional uninvolved skin, is the presence of granular deposits of immunoglobulin A (IgA) along the dermo-epidermal junction (DEJ) and at the tips of the dermal papillae. Recently, it has been documented that the autoantigen for deposited cutaneous IgA is epidermal transglutaminase (eTG, TG3)—an enzyme closely related, but not identical to the tissue transglutaminase (tTG, TG2) autoantigen-specific for CD [3]. IgA deposits in skin represent antibodies against gut tTG that cross-react with the highly homologous eTG by forming insoluble aggregates in the papillary dermis [4].

The pathophysiology of DH is closely related to that of CD and involves a complex interplay among genetic, environmental, and immune factors. Both diseases occur in gluten-sensitive individuals, heal with a gluten-free diet (GFD), and relapse on gluten challenge [5]. DH and CD share the same genetic background with a high frequency of human leukocyte antigen (HLA)-DQ2 and HLA-DQ8 haplotypes [6,7]. The majority of patients with DH exhibit morphologic small-bowel changes characteristics of CD, ranging from slight villous atrophy to increased density of intraepithelial lymphocytes [1,8]. However, overt enteropathy is reported in less than 10% of patients, and the gastrointestinal symptoms are usually absent or so mild that the DH patients are unaware of them [9]. Last but not least, patients with DH and CD often have the same associated autoimmune diseases, such as juvenile diabetes, hypothyroidism, pernicious anemia, and connective tissue disorders [5].

A hallmark of CD is the loss of tolerance to wheat gluten with enhanced production of various gluten-dependent autoantibodies, as a result from the gluten-induced small-bowel mucosal T-cell activation, which is the cornerstone in the pathogenesis of the celiac pathology [10]. These circulating CD-specific antibodies are widely used to diagnose GSE serologically before proceeding to small-bowel mucosal biopsies. Historically, among the first serum-based antibody tests introduced in CD diagnostics are the antigliadin antibody (AGA) [11,12], the gluten-dependent IgA-class R1-type reticulin (ARA) [13], and endomysial autoantibody (EMA) assays [14]. In 1997, Dieterich and co-workers identified TG2 as the autoantigen of CD [15]. As various TG2-based enzyme-linked immunosorbent assays (ELISA) became available, a new era in celiac disease case finding by serology began [16]. Later research has shown that TG2 was also the specific protein antigen in the ARA and EMA tests [17]. As a result of the constant development of serologic tests for CD, a new generation of assays detecting the presence of antibodies against deamidated gliadin peptides (DGPs) as antigens appeared [18,19]. The accurate diagnosis of DH is essential, similar to CD, as the disease requires a lifelong commitment to a GFD. It relies on few but essential specific criteria, including clinical, histologic, immunopathologic, and serologic celiac-related markers, the latter being detected in DH patients as well [2,20]. Perilesional biopsy with a specific DIF microscopy finding has remained the gold standard along with the presence of suggestive clinical picture and supportive serological results [21].

Furthermore, the predictive accuracy of serological tests depends on the disease prevalence in the population [22]. In this regard, it is of interest to analyze the performance of celiac-related tests in patients from different countries and origin. In a previous report of a series of 78 DH patients from Bulgaria, the prevalence of DH among other autoimmune blistering diseases was 7.45% with a minimum estimated incidence of 0.88 cases per million annually [23].

An early event in blister formation in DH is the accumulation of neutrophils in the papillary dermis, the upregulation of the adhesion molecules, and release of enzymes and inflammatory mediators causing basement membrane damage and subsequent clefting, which could also explain the typical distribution of skin lesions at sites of trauma [24]. Interleukin (IL)-17A is involved in the production of other pro-inflammatory cytokines and matrix metalloproteinases, as well as in the attraction of neutrophils implicated in the pathogenesis of DH [25]. However, the suggested hypothesis for the role of IL-17A in DH pathogenesis needs further investigation.

Our study aimed to explore comparatively the performance of a panel of celiac-related antibodies, such as anti-tTG, AGA, anti-DGP, anti-actin (AAA) antibodies, as well as cytokine IL-17A, in a cross-sectional study of a Bulgarian cohort of DH patients.

## 2. Material and Methods

### 2.1. Serum Samples

Sera from 26 newly diagnosed and untreated DH patients (mean age 53 ± 15 years; range 18–72 years) were collected before initiation of a gluten-free diet. All patients attended the Department of Dermatology, Aleksandrovska University Hospital, Sofia and provided written informed consent to participate in the study. The diagnosis of DH was based on (i) clinical presentation and (ii) presence of granular deposits of IgA in the papillary dermis by direct IF microscopy. Sera from 20 healthy individuals at mean age 31 ± 8 (range 21–42 years) served as controls. All sera were stored at −80 °C until assayed. Female-to-male ratio for DH patients was 1:1, and for the control group 1:1.2. Age and sex differences between the studied groups were considered as non-significant (*p* > 0.05). All patients and control subjects were found negative for other autoimmune disease markers (i.e., anti-nuclear antibodies, rheumatoid factor, and anti-neutrophil cytoplasmic antibodies).

This study was performed in accordance with the declaration of Helsinki Principles and approved by the Ethical Committee of the Medical University of Sofia, Bulgaria.

### 2.2. Immune Serology Testing

Sera taken from all DH patients and control subjects were analyzed by ELISA and immunoblotting (Line Blot) at the Laboratory of Clinical Immunology, University Hospital “St. Ivan Rilski,” Sofia.

#### 2.2.1. Immunoenzyme Testing

ELISA commercial kits were used to determine the following celiac-related antibodies and the pro-inflammatory cytokine IL-17A:anti-tTG antibodies (Anti-Tissue Transglutaminase Screen IgA + IgG, Orgentec Diagnostika GmbH, cut-off value > 10 U/mL);anti-DGP antibodies (Quanta Lite Celiac DGP Screen IgA + IgG, Inova Diagnostics, Inc., San Diego, USA, cut-off > 15 U/mL);AAA (Quanta Lite F-Actin IgA ELISA, Inova Diagnostics, Inc., San Diego, USA, cut-off > 20 U/mL);IL-17A (Human IL-17A ELISA kit, Diaclone, GenProbe, France, sensitivity < 2.3 pg/mL).

Analyses were performed following the manufacturers’ instructions.

#### 2.2.2. Immunoblot Testing

Anti-tTG, AGA, and ASCA were assessed in serum samples by performing line blot testing (Seraline® Zöliakie-3 IgG, Seramun Diagnostica GmbH, Germany). The assay strips were scanned with IvD-registered Seraline Scan software with hardware key (Seramun Diagnostica GmbH, Germany). The results were given as the relative value of intensity.

### 2.3. Statistical Analysis

Row data were evaluated statistically by the software package for statistical analysis (SPSS) v.19 (SPSS®, IBM 2009). We used descriptive, correlation, and receiver operating characteristics (ROC) curve analysis to evaluate the performance characteristics of the applied tests in diagnosing DH. Results are presented as mean ± SE (standard error) or number (%). Differences between the groups were assessed using unpaired Student’s T-test preceded by an evaluation of normality (Kolmogorov–Smirnoff test). The Mann–Whitney U-test was used where appropriate. A *P*-value of <0.05 was considered statistically significant.

## 3. Results

### 3.1. Serum Levels of the Celiac Disease-Related Autoantibodies and the Pro-Inflammatory Cytokine IL-17A

The mean ELISA values of the measured parameters in DH patients and the control group are presented on Figure 1 and Appendix A. The mean levels of anti-tTG and anti-DGP antibodies were significantly higher in DH patients compared to healthy controls (36.9 ± 20.3 IU/mL versus 2.1 ± 0.4 IU/mL, *p* = 0.02, and 40.7 ± 10.2 IU/mL versus 1.87 ± 0.68, *p* < 0.001, respectively). Similarly, the AAA titers significantly differed between both groups, being moderately higher in DH sera than in the healthy subjects (22.6 ± 3.9 IU/mL versus 9.1 ± 0.9 IU/mL, *p* = 0.05). There was a 60-fold increase in the concentrations of IL-17A in DH patients compared to control sera (5.3 ± 2.2 pg/mL versus 0.08 ± 0.07 pg/mL, *p* = 0.031) (Figure 1A).

The mean serum levels of the autoantibodies investigated by Line blot are also demonstrated (Figure 1 and Appendix A). There were significantly higher levels of anti-tTG and AGA antibodies in DH patients compared to healthy controls (0.88 ± 0.24 versus 0.08 ± 0.02, *p* = 0.003, and 0.98 ± 0.31 versus 0.25 ± 0.08, *p* = 0.030, respectively). In contrast, no differences were found in the mean levels of ASCA within the studied groups (Figure 1B).

### 3.2. Performance Characteristics of the Celiac-Related Antibodies Tested in DH Patients

The results of the performance of anti-tTG, anti-DGP antibodies, AAA, and AGA, assessed by ELISA and line blot are shown in Table 1. Antibodies against tTG were found in 11 (42.3%) (IgA + IgG, ELISA) and 12 (46%) (IgG, line blot) patients with DH. Half of the DH patients had AGA IgG (Line blot) in their sera, and 12 (46.4%) were positive for anti-DGP antibodies. The smallest number of patients—9 (34.7%) were found positive for AAA (ELISA).

None of the control sera were tested positive for anti-tTG (ELISA and blot), AAA or AGA, whereas one subject showed positive results for anti-DGP. This defined a specificity of 100% in distinguishing DH from healthy individuals for the test systems applied in our study, excluding anti-DGP antibodies, which exerted a specificity of 95%.

Positive predictive values (PPV) for all tests were 100%, except for anti-DGP—90.9%. The negative predictive values (NPV) of the test remained slightly above 50%, and the highest NPV was observed for AGA (60%) and anti-tTG (Line blot) (59%).

### 3.3. ROC Curve Analysis of the Celiac Disease-Related Antibodies and IL-17A in DH Patients

The ROC curve analyses of the ELISA tests revealed the best performance of anti-DGP antibodies (AUC 0.939, *p* < 0.001), followed by anti-tTG antibodies testing (AUC 0.864, *p* = 0.002) (Appendix A). We did not find significant AUC for AAA. According to IL-17 serum levels, our results demonstrated excellent performance of the test (AUC 0.811, *p* < 0.05) (Figure 2A). From the celiac-related antibodies assessed by line blot, anti-tTG testing alone had significant AUC of 0.734, while the other tests showed unsatisfactory performance (Figure 2B).

### 3.4. Correlation between Tests

The results of all tests showed good to excellent correlation to each other (r = 0.5÷0.9, *p* < 0.01) (Table 2). The strongest correlations were established for the following pairs of antibodies, all of them assessed by ELISA: anti-tTG—IL-17A (r = 0.938, *p* < 0.001), anti-tTG – anti-DGP (r = 0.894, *p* < 0.001), and anti-tTG—AAA (r = 0.863, *p* = 0.001). In comparison, the correlation between anti-DGP antibodies and IL-17A was evaluated as a weak one (r = 0.452, *p* = 0.031). Anti-tTG ELISA levels moderately correlated with anti-tTG assessed by line blot (r = 0.520, *p* = 0.003) (Table 2).

## 4. Discussion

Growing evidence shows that patients with DH may possess most of the specific autoantibodies that can be found in patients with CD, including circulating autoantibodies against gliadin, tTG, and DGP [1]. On the other hand, conflicting results were obtained by the use of the anti-DGP ELISA for detecting gluten enteropathy in DH patients. Previously reported sensitivities for IgA anti-DGP antibodies vary from 46% to 78% [20,26]. In this study, the relative sensitivities and specificities of a panel of CD-related autoantibodies in Bulgarian patients with DH were compared with the reactivities of control healthy subjects. We included conventional celiac-related antibodies—anti-tTG, anti-DGP, and AGA, as well as AAA, the latter being used for non-invasive evaluation of villous atrophy. ASCA were tested along with other antibodies due to the presence of coated Mannan on the Line blot. Moreover, we were interested in assessing the serum levels of IL-17A in DH patients. We chose not to compare EMA with the other autoantibodies in our celiac-related panel due to the subjective semiquantitative nature of EMA testing that is not easy to standardize.

All investigated celiac-related antibodies—anti-tTG, anti-DGP, and AGA, independent of the used method (ELISA or Line blot), were significantly higher in the DH group compared to the healthy controls. Nevertheless, the sensitivity and specificity of the applied tests were acceptable. We found that 42.3% of our DH patients were positive for anti-tTG (IgA + IgG) assessed by ELISA. When we tested the serum samples for IgG anti-tTG by line blot, we found a higher sensitivity of 46%. Half of the DH patients had IgG AGA (Line blot) in their serum samples, and 46.4% had anti-DGP (ELISA) antibodies. We also defined the specificity of 100% for anti-tTG (ELISA and line blot), AAA, and AGA in discriminating DH from healthy persons, as well as a specificity of 95% for anti-DGP antibodies. These results are in accordance with other studies, demonstrating sensitivity ranges between 47% and 100% and specificity ranging 90% to 100% for celiac-related antibodies in patients with DH [9,27,28,29,30,31,32]. PPVs for all tests were 100%, except for anti-DGP, which was 90.9% due to one positive healthy individual. Unfortunately, the NPVs of the tests remained slightly above 50%, and the highest NPV was observed for AGA (60%) and anti-tTG (59%) assessed by line blot. However, during the last decade, only a few studies updated this information. Thus, our results contribute to previously published literature data.

Comparing tests by the ROC curve analyses, the best performance was revealed for anti-DGP antibodies, followed by anti-tTG (ELISA) testing and anti-tTG (Line blot) antibodies. Although the specificity of AGA was 50%, the AUC of 0.600 was non-significant and therefore, unreliable.

Among all celiac-related serological tests, IgA anti-tTG antibodies have been considered the most sensitive and specific ones that should be tested in patients with DH symptoms [1]. Since some patients with DH or CD may have selective IgA deficiency, we chose the dual IgG/IgA test system to exclude false-negative results. [27,33]. In our study, the performance of anti-DGP in diagnosing DH was shown to be superior to the anti-tTG antibodies. In previous comparative studies among DH patients, the sensitivity and specificity of anti-DGP were either lower than those of anti-tTG and EMA, similar, or superior to them [34], as it is in the present study. The possible explanations for such discrepancies lie in the fact that anti-DGP and AGA, which are directed against deamidated gliadin peptides and whole gliadin peptide, respectively, are related to the presence of intestinal damage, whereas antibodies against the converting enzyme tTG are linked not only to mucosal but also to skin lesions as well [34]. However, current knowledge has shown that the available serologic armamentarium lacks sensitivity when used in patients with mild or minor enteropathy [35,36]. The similarity of DH and CD related to the enteropathy makes DH a fascinating model of skin CD, where papulovesicular and pruritic rash can be concomitant with a broad spectrum of intestinal damage varying from normal structure to villous atrophy [37]. However, DH is the second gluten-sensitive disorder exhibiting varied histological damage where one can assess the performance of the celiac serology [34]. In the present study, we chose to assess by ELISA anti-tTG and anti-DGP antibodies of both IgA and IgG subclasses. The results obtained allowed us to conclude that the combination of both isotypes of anti-DGP assays has higher specificity than IgA anti-tTG.

There is an insufficient number of investigations regarding anti-actin testing in DH patients. Of the 26 patients with DH in our study, nine were positive for AAA. However, no significant differences were found in the serum levels of AAA in DH patients and healthy controls. Serum levels of IgG AAA were assessed by ELISA in a single study on a series of 10 adult Romanian DH patients. The authors documented sensitivity and specificity of 33.3% and 100%, respectively, for AAA in DH patients [38]. Our results also showed that the AUC for AAA was unacceptable and therefore not reliable for the DH diagnosis.

We did not find differences in the mean levels of ASCA within the studied groups. Although ASCA have been reported to be positive in about 30% of CD patients [39], which was also confirmed by us in a cohort of Bulgarian CD patients [40], there were no data regarding ASCA in DH patients available so far.

Concerning the IL-17A, in a single study Zebrowska et al. documented significantly higher expression of this cytokine in the epidermis (perilesional skin) and the serum of DH and bullous pemphigoid patients, compared to the control group [41]. We also detected 60-fold higher concentrations of IL-17A in DH patients compared to healthy controls (*p* = 0.031). Two studies provided data for the involvement of IL-17A in DH pathogenesis. Juczynska et al. demonstrated increased expression of JAK/STAT proteins in skin lesions in patients with DH and bullous pemphigoid in comparison to perilesional skin and control group [42]. They suggested that pro-inflammatory cytokine network and induction of inflammatory infiltrate in tissues can contribute to the pathogenesis of skin lesions in both diseases. Surprisingly, serum IL-17 demonstrated excellent performance in our study (AUC 0.811, *p* = 0.008), which could be of benefit for the clinical practice.

We found good to excellent correlation (r = 0.5 ÷ 0.9, *p* < 0.01) between the tests. The strongest correlations were established for the following pairs: anti-tTG (ELISA)—IL-17A, anti-tTG (ELISA)—anti-DGP antibodies, and anti-tTG (ELISA)—AAA. These results suggested a good coincidence between the different tests in diagnosing DH. There was a moderate correlation between anti-tTG antibodies estimated by ELISA and by line blot, which is not encouraging regarding the interchangeability between the two methods for anti-tTG detection. Previous studies showed similar correlations between celiac-related antibodies in patients with GSE [40].

This study has some limitations. The relatively small size of the study population might have affected the significance of the results. The lack of data on anti-TG3 is another weak point of the present work. We assume that further research involving a larger number of DH patients and newly emerging test systems for detection of other transglutaminase antibodies (TG3 and/or TG6) would clarify the findings presented in the current study and may have a significant impact on the clinical practice.

## 5. Conclusions

Serologic tests are important noninvasive screening tool among symptomatic patients with clinical suspicion of DH that can help select patients for diagnostic DIF analysis. Furthermore, such tests are helpful in the resolution of ambiguous and false-negative DIF results. The usability of serologic DH tests is defined by their sensitivity and specificity, which are quite variable based on current data. This is due to scarcity of data from limited populations.

In this respect, serologic testing with a panel of celiac-related antibodies, rather than individual ones, may be successfully employed to support the diagnosis of DH. In our study, the performance of anti-DGP in diagnosing DH was shown to be superior to that of anti-tTG antibodies. In addition, the best performance (ROC curve analysis) was revealed for anti-DGP antibodies followed by anti-tTG ELISA. This is the first such study among Bulgarian patients and hopefully more will follow. Further studies among different populations are needed in order to improve evidence-based results and to decrease interpolation of data.

## Figures and Tables

**Figure 1 medicina-55-00136-f001:**
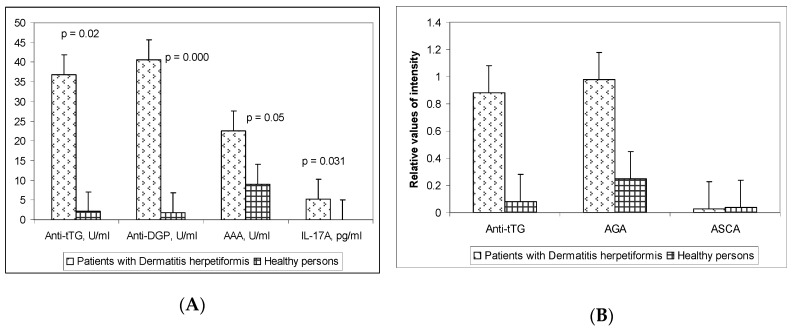
Mean serum levels of anti-tTG antibodies, anti-DGP, anti-actin antibodies, and IL-17A in the study groups, assessed by (**A**) ELISA and (**B**) line blot.

**Figure 2 medicina-55-00136-f002:**
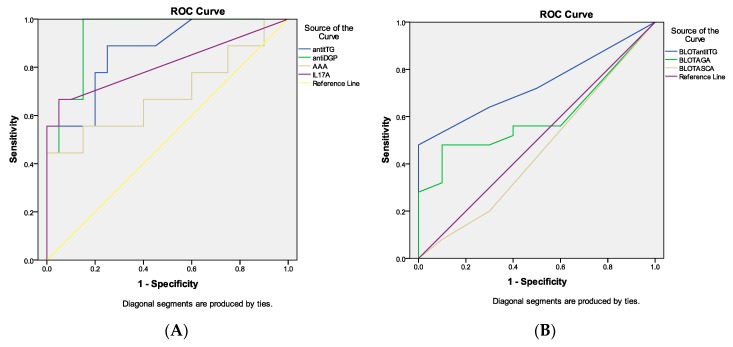
Receiver operating characteristics (ROC) curve analysis of the tested parameters, assessed by (**A**) ELISA and (**B**) line blot.

**Table 1 medicina-55-00136-t001:** Performance characteristics of anti-tTG antibodies, anti-DGP antibodies, AAA, and AGA, assessed by ELISA and line blot in Dermatitis Herpetiformis patients.

	Anti-tTG IgA + IgG (ELISA)	Anti-DGP IgA + IgG (ELISA)	AAA IgG (ELISA)	Anti-tTG IgG (Line Blot)	AGA IgG (Line Blot)
Sensitivity	42.3%	46.4%	34.7%	46%	50%
Specificity	100%	95%	100%	100%	100%
PPV *	100%	90.9%	100%	100%	100%
NPV **	57%	57.1%	54.1%	59%	60%

* PPV, positive predictive value; ** NPV, negative predictive value.

**Table 2 medicina-55-00136-t002:** Correlation between tests. Results are presented as Pearson’s coefficient (r) and significance (p).

	Anti-tTG (ELISA)	Anti-DGP (ELISA)	AAA (ELISA)	IL-17A (ELISA)	Anti-tTG (Line Blot)	AGA (Line Blot)
Anti-tTG (ELISA)		r = 0.894 *p* < 0.001	r = 0.863 *p* = 0.001	r = 0.938 *p* < 0.001	r = 0.520 *p* = 0.003	r = 0.507 *p* = 0.076
	anti-DGP (ELISA)		r = 0.502 *p* = 0.009	r = 0.452 *p* = 0.031	r = 0.532 *p* = 0.001	r = 0.346 *p* = 0.038
		AAA (ELISA)		r = 0.692 *p* < 0.001	r = 0.112 *p* = 0.500	r = 0.221 *p* = 0.186
			IL-17A(ELISA)		r = 0.079 *p* = 0.676	r = −0.222 *p* = 0.238
				Anti-tTG (Line blot)		r = 0.678 *p* < 0.001
					AGA (Line blot)	

All Pearson’s coefficients were calculated by bivariate correlation, except for the line blot results where the Spearman coefficient was calculated via Chi-square test.

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
