# Peer review of "Celiac-Related Autoantibodies and IL-17A in Bulgarian Patients with Dermatitis Herpetiformis: A Cross-Sectional Study"

_medicina, 2019, doi:10.3390/medicina55050136_

Round 1
Reviewer 1 Report
It is a work, original, very well presented, which I think will have many references in the future
ABSTRACT
Appropriate, a perfect summary of the work in terms of methods, results and conclusion
INTRODUCTION
A good summary of the current state of dermatitis herpetiformis, with updated citations
MATERIAL and METHODS
Exemplary description It is clearly indicated that the possible confusion by age and sex is not significant
RESULTS
Very well presented. High quality of tables and figures
DISCUSSION
The advantages and disadvantages of this novel approach to the diagnosis and follow-up of the disease are clearly presented
CONCLUSIONS
correct, according to the results and discussion
Author Response
Thank you for your valuable comments and evaluation of the manuscript. Our research question was devoted to Bulgarian patients with DH and their assessment with celiac-related antibodies and the cytokine IL-17A. In our cross-sectional study, we aimed to demonstrate that serologic testing of a panel of celiac related-antibodies could be employed with advantages in the diagnosing process of DH patients and for routine monitoring for gastrointestinal complications typical for the gluten-sensitive enteropathy.

Reviewer 2 Report
In the paper entitled:" Celiac-related autoantibodies and IL-17A in Bulgarian patients with
dermatitis herpetiformis: a cross-sectional study", the Authors investigated several deliac-specific antibodies together with IL-17A serum levels in a cohort of 26 patients with dermatitis herpetiformis (DH) and in healthy controls. Although the topic is interesting, the originality of the paper is questionable, since several papers have tested the same antibodies in patients with DH. Moreover, even IL-17A has been previously tested in patients with DH. Moreover, no novel data come out from this study, since it is well known that anti-tTG and anti-DGP antibodies work quite well in serologic testing of patients with DH, having a high specificity (and a moderate sensitivity), while they are usually absent in healthy controls. Accordingly, even the conclusions are not novel. Therefore, probably the paper would be more suitable for the publication in a more concise form (i.e. a letter to the Editor). Other points are discuussed below:
- In general, the paper is too long. In particular, the introduction and the discussion should be shortened. Despite this, the better way to show these results will be a letter to the Editor.
- The number of the patients included in the study is quite low, and this may affect the significance of the results. This should be clearly stated as a limitation of the study.
- Anti-epidermal TG antibodies are considered the most important serologic marker for patients with DH. Therefore, it would be interesting to test such antoibodies in the cohort of the 26 patients included into this study. Otherwise, the Authors should explain the reasons for which they did not perform such an analysis.
- Figure 1 and table 1 are redundant, as are figure 2 and table 3. Therefore, I suggest to delete the tables 1 and 3.
- The conclusions are not novel and in part questionable, and should be rewritten. In fact, the usefulness of testing for anti-DGP and anti-tTG antibodies patients with DH with a typical DIF results is not clear. In fact, for the diagnosis, DIF is the gold standard and with a positive DIF no more diagnostic tests would be required. By contrast, in patients with a negative DIF, but with a high suspicion of DH, serologic testing may be helpful to exclude DH or CD in the case that DIF was false negative.
Moreover, since patients with a confirmed DH should also be considred as having celiac disease, it would not be correct to monitor for gastroenterological complications only those that resulted positive to serologic testing.
Author Response
In the paper entitled:" Celiac-related autoantibodies and IL-17A in Bulgarian patients with
dermatitis herpetiformis: a cross-sectional study", the Authors investigated several deliac-specific antibodies together with IL-17A serum levels in a cohort of 26 patients with dermatitis herpetiformis (DH) and in healthy controls. Although the topic is interesting, the originality of the paper is questionable, since several papers have tested the same antibodies in patients with DH. Moreover, even IL-17A has been previously tested in patients with DH. Moreover, no novel data come out from this study, since it is well known that anti-tTG and anti-DGP antibodies work quite well in serologic testing of patients with DH, having a high specificity (and a moderate sensitivity), while they are usually absent in healthy controls. Accordingly, even the conclusions are not novel. Therefore, probably the paper would be more suitable for the publication in a more concise form (i.e. a letter to the Editor).
Ø Thank you for the constructive critic. We agree with the referee that the used celiac-related antibodies were introduced for DH diagnostics before, as well as IL-17A was also investigated. However, to the best of our knowledge, there is no such study among the Bulgarian population. In this way, it contributes to the understanding in the field, and the results could be used in future systematic reviews and meta-analysis.
Ø According to other comments – we confirmed the usefulness of these antibodies for diagnosing of DH.
Ø Besides, we also compared different type of immunological tests (ELISA, immunoblot) for these antibodies to provide information regarding which test is better in the laboratory and clinical settings.
Ø The conclusions are based on the obtained results.
Ø However, the proposed manuscript belongs more to the original article. Thus, we prefer to keep this article type for publishing. We believe that our paper meets the criteria for publishing in the special issue about celiac disease and related conditions.
Other points are discuussed below:
- In general, the paper is too long. In particular, the introduction and the discussion should be shortened. Despite this, the better way to show these results will be a letter to the Editor.
Ø Thank you for the note. Indeed, we extend the introduction and discussion section. The reason for this was to reveal what is already known about this topic, to discuss the results from different angles and place them into context without being over-interpreted. Once again we insist that our manuscript fits mostly the original article type.
- The number of the patients included in the study is quite low, and this may affect the significance of the results. This should be clearly stated as a limitation of the study.
Ø We agree with the referee that the number of patients included is relatively small. We have to notice, however, that DH is a rare disease, exceptionally rare are the newly diagnosed patients which we recruited in our study.
Ø We have revised the text where the limitations were outlined clearly (page 9, lines 303-306). We believe that these limitations of the study are not fatal, but they are opportunities to inform future research.
- Anti-epidermal TG antibodies are considered the most important serologic marker for patients with DH. Therefore, it would be interesting to test such antoibodies in the cohort of the 26 patients included into this study. Otherwise, the Authors should explain the reasons for which they did not perform such an analysis.
Ø It is true that anti-epidermal TG antibodies are considered the essential serologic marker for patients with DH. Unfortunately, the funding body for our project covered a limited amount of expenses. Thus, we could not manage to include testing of anti-eTG for the mentioned project. However, we keep all the serum samples of the patients frozen, and we are planning to further extend our research in investigating additional serological markers for DH patients, including anti-eTG antibodies.
- Figure 1 and table 1 are redundant, as are figure 2 and table 3. Therefore, I suggest to delete the tables 1 and 3.
Ø The referee is right to point out that there is some similarity between the respective figures and tables. However, tables provide additional information which could be useful for other researchers and clinicians, or meta-analysis.
- The conclusions are not novel and in part questionable, and should be rewritten. In fact, the usefulness of testing for anti-DGP and anti-tTG antibodies patients with DH with a typical DIF results is not clear. In fact, for the diagnosis, DIF is the gold standard and with a positive DIF no more diagnostic tests would be required. By contrast, in patients with a negative DIF, but with a high suspicion of DH, serologic testing may be helpful to exclude DH or CD in the case that DIF was false negative.
Ø We support the referee’s assertion that the conclusions are not novel. We did not intend for novelty in findings, but to answer the aim of the study. Moreover, the outcomes are supported by references and our results.
Ø We have rewritten the conclusion in regards to the referee`s comments (page 9, lines 317-321).
Moreover, since patients with a confirmed DH should also be considred as having celiac disease, it would not be correct to monitor for gastroenterological complications only those that resulted positive to serologic testing.
Ø Thank you for the valuable comment. We agree that all patients should be monitored gastroenterologically for complications. We have corrected the mentioned issue.

Round 2
Reviewer 2 Report
The Authors made some of the suggested corrections, including the limitations of the study and changing the conclusions. Accordingly, I suggest to change the paragraph on the limitations of the study deleting the following sentences: "We have to notice, however, that DH is a rare disease, especially the newly diagnosed patients which we recruited in our study. We believe that these limitations of the study are not fatal, but they are opportunities to inform future research."
Moreover, please state clearly that the lack of data on anti-TG3 is a major limitation.
In general, however, the manuscript did not changed a lot after the revision and most of the issues still remain.
Author Response
Dear reviewer,
Thank you once again for your efforts to review our revised paper medicina-450263 entitled Celiac-related autoantibodies and IL-17A in Bulgarian patients with dermatitis herpetiformis: a cross-sectional study.
We understand that you are not satisfied with the previous changes made to our manuscript. Therefore, we provide an extensive revision that we hope you will find satisfactory.
Open Review
(x) I would not like to sign my review report
( ) I would like to sign my review report
English language and style
( ) Extensive editing of English language and style required
(x) Moderate English changes required
( ) English language and style are fine/minor spell check required
( ) I don't feel qualified to judge about the English language and style
Ø |
Yes | Can be improved | Must be improved | Not applicable | |
Does the introduction provide sufficient background and include all relevant references? | (x) | ( ) | ( ) | ( ) |
Is the research design appropriate? | (x) | ( ) | ( ) | ( ) |
Are the methods adequately described? | (x) | ( ) | ( ) | ( ) |
Are the results clearly presented? | (x) | ( ) | ( ) | ( ) |
Are the conclusions supported by the results? | (x) | ( ) | ( ) | ( ) |
Comments and Suggestions for Authors
Ø With regards to the moderate English changes required: final grammar and spelling changes were made by a professional proofreader.
The Authors made some of the suggested corrections, including the limitations of the study and changing the conclusions. Accordingly, I suggest to change the paragraph on the limitations of the study deleting the following sentences: "We have to notice, however, that DH is a rare disease, especially the newly diagnosed patients which we recruited in our study. We believe that these limitations of the study are not fatal, but they are opportunities to inform future research."
Moreover, please state clearly that the lack of data on anti-TG3 is a major limitation.
Ø We have deleted the suggested paragraph. Instead, we describe the limitation as the referee proposed.
In general, however, the manuscript did not changed a lot after the revision, and most of the issues still remain.
Ø We have made additional changes according to the previous comments of the referee, as follows:
o Regarding the novelty of the results, we have provided our arguments that our paper is first to describe this problem in Bulgarian DH patients. We still insist on the form of the paper to remain an original article. We believe that these issues are resolved.
o Regarding the suggestion to shorten the paper – we have reduced the introduction, discussion and results sections.
o Regarding the number of DH patients recruited and lack of investigation of anti-TG3, we have added these as limitations to our study as suggested by the reviewer.
o Regarding the redundant figures and tables, we have deleted tables 1 and 3, as the referee suggested. We have added some of the information from them in the main text. The two tables were added as supplementary materials to the manuscript.
o Regarding the conclusion, we revised it substantially in order to present the key points of our results.
o Regarding the gastroenterological follow-up of the DH patients, we have added this paragraph in the last revision. We consider this as a resolved issue.

Round 3
Reviewer 2 Report
The manuscript has improved